# A GNN-Guided Predict-and-Search Framework for Mixed-Integer Linear Programming

**Qingyu Han**[1,2†], **Linxin Yang**[1,3†], **Qian Chen**[1,4], **Xiang Zhou**[5], **Dong Zhang**[5], **Akang Wang**[1*], **Ruoyu Sun**[6,3*], **Xiaodong Luo**[1,3]

[1] Shenzhen Research Institute of Big Data, China

[2] Shandong University, China

[3] School of Data Science, The Chinese University of Hong Kong, Shenzhen, China

[4] School of Science and Engineering, The Chinese University of Hong Kong, Shenzhen, China

[5] Huawei, China

[6] Shenzhen International Center For Industrial and Applied Mathematics, Shenzhen Research Institute of Big Data, China

## Abstract

Mixed-integer linear programming (MILP) is widely employed for modeling combinatorial optimization problems. In practice, similar MILP instances with only coefficient variations are routinely solved, and machine learning (ML) algorithms are capable of capturing common patterns across these MILP instances. In this work, we combine ML with optimization and propose a novel predict-and-search framework for efficiently identifying high-quality feasible solutions. Specifically, we first utilize graph neural networks to predict the marginal probability of each variable, and then search for the best feasible solution within a properly defined ball around the predicted solution. We conduct extensive experiments on public datasets, and computational results demonstrate that our proposed framework achieves $51.1\%$ and $9.9\%$ performance improvements to MILP solvers SCIP and Gurobi on primal gaps, respectively.

## 1 Introduction

*Mixed-integer linear programming* is one of the most widely used techniques for modeling *combinatorial optimization* problems, such as production planning (Pochet & Wolsey, 2006; Chen, 2010), resource allocation (Liu & Fan, 2018; Watson & Woodruff, 2011), and transportation management (Luathep et al., 2011; Schöbel, 2001). In real-world settings, MILP models from the same application share similar patterns and characteristics, and such models are repeatedly solved without making uses of those similarities. ML algorithms are well-known for its capability of recognizing patterns (Khalil et al., 2022), and hence they are helpful for building optimization algorithms. Recent works have shown the great potential of utilizing learning techniques to address MILP problems. The work of (Bengio et al., 2021) categorized ML efforts for optimization as (i) *end-to-end learning* (Vinyals et al., 2015; Bello* et al., 2017; Khalil et al., 2022), (ii) *learning to configuring algorithms* (Bischl et al., 2016; Kruber et al., 2017; Gasse et al., 2022) and (iii) *learning alongside optimization* (Gasse et al., 2019; Khalil et al., 2016; Gupta et al., 2020). In this work, for the sake of interest, we focus on the *end-to-end* approach. While such an approach learns to quickly identify high-quality solutions, it generally faces the following two challenges:

(I) **high sample collection cost**. The supervised learning task for predicting solutions is to map from the instance-wise information to a high-dimensional vector. Such a learning task becomes computationally expensive since it necessitates collecting a considerable amount of optimal solutions (see, e.g., Kabir et al. (2009)).

(II) **feasibility**. Most of the end-to-end approaches directly predict solutions to MILP problems, ignoring feasibility requirements enforced by model constraints (e.g. Yoon (2022); Nair

---

†Equal first authorship    *Corresponding authors

et al. (2020)). As a result, the solutions provided by ML methods could potentially violate constraints.

We propose a novel *predict-and-search* framework to address the aforementioned challenges. In principle, end-to-end approaches for MILP problems require the collection of abundant optimal solutions. However, collecting so many training samples is computationally prohibitive since obtaining optimal solutions is excessively time-consuming. We consider to approximate the distribution by properly weighing high-quality solutions with their objective values. This reduces the cost of sample collections by avoiding gathering optimal solutions as mentioned in challenge (**I**). Regarding challenge (**II**), we implement a *trust region* inspired algorithm that searches for near-optimal solutions within the intersection of the original feasible region and a properly defined ball centered at a prediction point. The overall framework is outlined in Figure 1. The commonly used *end-to-*

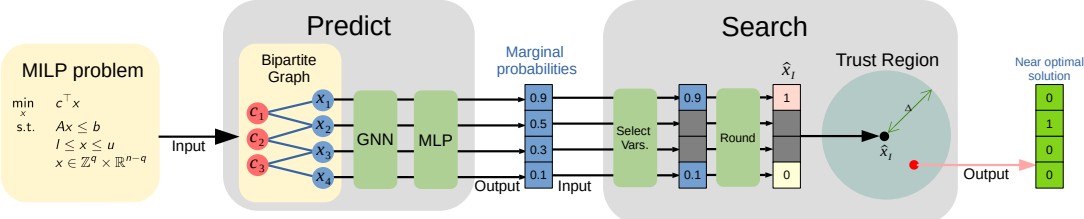

Figure 1: Our approach first predicts a marginal probability of each variable utilizing a *graph neural network* (GNN) with graph convolution and *multi-layer perceptron* modules, and then searches for near optimal solutions to the original MILP problem within a well defined trust region.

*end* approaches usually fix variables directly based on information from the prediction step, such as in Nair et al. (2020) and Yoon (2022). However, such approaches could lead to sub-optimal solutions or even infeasible sub-problems. Rather than forcing variables to be fixed, our search module looks for high-quality solutions within a subset of the original feasible region, which allows better feasibility while maintaining optimality.

The distinct contributions of our work can be summarized as follows.

- We propose a novel predict-and-search framework that first trains a GNN to predict a marginal probability of each variable and then constructs a trust region to search for high-quality feasible solutions.
- We demonstrate the ability of our proposed framework to provide equivalently good or better solutions than fixing-based end-to-end approaches.
- We conduct comprehensive computational studies on several public benchmark datasets and the computational results show that our proposed framework achieves $51.1\%$ and $9.9\%$ smaller primal gaps than state-of-the-art general-purpose optimization solvers SCIP and Gurobi, respectively.

We make our code publicly available at `https://github.com/sribdcn/Predict-and-Search_MILP_method`.

## 2 PRELIMINARIES

Given a vector $v \in \mathbb{R}^n$ and an index set $I \subseteq \{1, 2, ..., n\}$, let $v_I \in \mathbb{R}^{|I|}$ denote a subvector that corresponds to the index set $I$.

### 2.1 MIXED-INTEGER LINEAR PROGRAMMING

MILP techniques are used to model combinatorial optimization problems, and an MILP instance can be formulated as: $\min_{x \in D} c^\top x$, where $D \equiv \{x \in \mathbb{Z}^q \times \mathbb{R}^{n-q} : Ax \leq b, l \leq x \leq u\}$ denotes the

feasible region. There are $n$ variables, with $c, l, u \in \mathbb{R}^n$ being their objective coefficients, lower and upper bounds, respectively. Without loss of generality, the first $q$ variables are discrete. $A \in \mathbb{R}^{m \times n}$ denotes the coefficient matrix while $b \in \mathbb{R}^m$ represents the right-hand-side vector. For convenience, let $M \equiv (A, b, c, l, u, q)$ denote an MILP instance. For the sake of interest, we consider each discrete variable to be binary, i.e. $x_i \in \{0, 1\}$ for $1 \leq i \leq q$. Furthermore, incorporating continuous variables into consideration will not invalidate our proposed methodology, hence in what follows we consider a pure binary integer programming problem.

## 2.2 NODE BIPARTITE GRAPH

Gasse et al. (2019) proposed a bipartite graph representation for MILP problems. Specifically, let $G \equiv (\mathcal{V}, \mathcal{E})$ denote a bipartite graph, where $\mathcal{V} \equiv \{v_1, ..., v_n, v_{n+1}, ..., v_{n+m}\}$ denotes the set of $n$ variable nodes and $m$ constraint nodes, and $\mathcal{E}$ represents the set of edges that only connect between nodes of different types. Variable nodes and constraint nodes are individually associated with the instance information e.g. degrees and coefficients. The adjacency matrix $\mathcal{A}$ is a $|\mathcal{V}| \times |\mathcal{V}|$ matrix that represents the connectivity of $G$, as defined below:

$$\mathcal{A} \equiv \begin{bmatrix} 0 & C^\top \\ C & 0 \end{bmatrix} \text{ where } C_{i,j} = \mathbb{1}_{A_{i,j} \neq 0}.$$

## 2.3 GRAPH NEURAL NETWORKS

Let $N(v_i) \equiv \{v_j \in \mathcal{V} : \mathcal{A}_{i,j} \neq 0\}$ denote the set of neighbors of node $v_i$. We construct a $k$-layer GNN as follows:

$$h_{v_i}^{(k)} \equiv f_2^{(k)} \left( \left\{ h_{v_i}^{(k-1)}, f_1^{(k)} \left( \left\{ h_u^{(k-1)} : u \in N(v_i) \right\} \right) \right\} \right),$$

where function $f_1^{(k)}$ aggregates the feature information over the set of neighboring nodes and function $f_2^{(k)}$ combines the nodes' hidden features from iteration $(k-1)$ with the aggregated neighborhood features, and $h_{v_i}^{(k-1)}$ denotes the hidden state of node $v_i$ in the $(k-1)^{th}$ layer. Initially, $h_{v_i}^{(0)}$ is the output of an embedding function $g(\cdot)$. Note that the bipartite graph associated with an MILP instance does not include edges between variable nodes as well as constraint nodes.

## 2.4 TRUST REGION METHOD

The trust region method is designed for solving non-linear optimization problems as follows:

$$\min_{x \in H} f(x), \tag{1}$$

where $H \subseteq \mathbb{R}^n$ denotes a set of feasible solutions. The main idea is to convert the originally difficult optimization problem into a series of simple local search problems. Specifically, it iteratively searches for trial steps within the neighborhood of the current iterate point (Yuan, 2015). The trial step $d_k$ is obtained by solving the following trust region problem:

$$\begin{aligned} \min_{d \in H_k} \quad & \widetilde{f}_k(d), \\ \text{s.t.} \quad & \|d\|_{W_k} \leq \Delta_k \end{aligned} \tag{2}$$

where $\widetilde{f}_k(d)$ is an approximation of the objective function $f(x_k + d)$, $H_k$ denotes a shifted $H - x_k$ of the set $H$, $\|.\|_{W_k}$ is a norm of $\mathbb{R}^n$, and $\Delta_k$ denotes the radius of the trust region. After solving the problem in the $k^{th}$ iteration, $x_{k+1}$ is updated to $x_k + d_k$ or $x_k$ accordingly. Then, new $\|.\|_{W_{k+1}}$ and $\Delta_{k+1}$ are selected along with a new approximation $\widetilde{f}_{k+1}(d)$.

## 3 PROPOSED FRAMEWORK

The supervised learning task in end-to-end approaches trains a model to map the instance-wise information to a high-dimensional vector. Ding et al. (2020) first utilizes a GNN to learn the values of

"stable" variables that stay unchanged across collected solutions, and then searches for optimal solutions based on learned information. However, such a group of variables does not necessarily exist for many combinatorial optimization problems. Another approach is to learn *solution distributions* rather than directly learning *solution mapping*; Li et al. (2018) predicts a set of probability maps for variables, and then utilizes such maps to conduct tree search for generating a large number of candidate solutions. Their work designed ML-enhanced problem-specific tree search algorithm and achieved encouraging results, however, it is not applicable to general problems. Nair et al. (2020) proposed a more general approach that first learns the conditional distribution in the solution space via a GNN and then attempts to fix a part of discrete variables, producing a smaller MILP problem that is computationally cheap to solve. However, fixing many discrete variables might lead to an infeasible MILP model.

To alleviate these issues, we adopt the trust region method and design a novel approach that searches for near-optimal solutions within a properly defined region. Specifically, we propose a predict-and-search framework that: (i) utilizes a trained GNN model to predict marginal probabilities of all binary variables in a given MILP instance; (ii) searches for near-optimal solutions within the trust region based on the prediction.

## 3.1 PREDICT

In this section, our goal is to train a graph neural network using supervised learning to predict the conditional distribution for MILP instances. To this end, we introduce the conditional distribution learning. On this basis, we present the training label in the form of a vector, i.e. marginal probabilities.

### 3.1.1 DISTRIBUTION LEARNING

A probability distribution learning model outputs the conditional distribution on the entire solution space of an MILP instance $M$. A higher conditional probability is expected when the corresponding solution is more likely to be optimal. Nair et al. (2020) proposed a method to construct the conditional distribution with objective values via energy functions, and for a solution $x$, the conditional probability $p(x; M)$ can be calculated by:

$$p(x; M) \equiv \frac{\exp(-E(x; M))}{\sum_{x'} \exp(-E(x'; M))}, \quad \text{where } E(x; M) \equiv \begin{cases} c^\top x & \text{if } x \text{ is feasible,} \\ +\infty & \text{otherwise.} \end{cases} \quad (3)$$

This implies that an infeasible solution is associated with a probability of $0$, while the optimal solution has the highest probability value. It's worth noting that each instance corresponds to one distribution.

For the collected dataset $\left\{ \left( M^i, \ L^i \right) \right\}_{i=1}^{N}$, $L^i \equiv \left\{ x^{i,j} \right\}_{j=1}^{N_i}$ denotes the set of $N_i$ feasible solutions to instance $M^i$. The probability of each solution in the dataset can be calculated by Equation (3). In general, the distance between two distributions can be measured by Kullback-Leibler divergence. Thus, the loss function for the supervised learning task is defined as:

$$L(\theta) \equiv -\sum_{i=1}^{N} \sum_{j=1}^{N_i} w^{i,j} \log P_\theta \left( x^{i,j}; M^i \right), \quad \text{where } w^{i,j} \equiv \frac{\exp \left( -c^{i\top} x^{i,j} \right)}{\sum_{k=1}^{N_i} \exp \left( -c^{i\top} x^{i,k} \right)}. \quad (4)$$

$P_\theta \left( x^{i,j}; M^i \right)$ denotes the prediction from the GNN denoted as $F_\theta$ with learnable parameters $\theta$. The conditional distribution of an MILP problem can be approximated by a part of the entire solution space. Consequently, the number of samples to be collected for training is significantly reduced.

### 3.1.2 WEIGHT-BASED SAMPLING

To better investigate the learning target and align labels with outputs, we propose to represent the label in a vector form. With the learning task specified in Equation (4), a new challenge arises: acquiring solutions through high-dimensional sampling is computationally prohibitive. A common technique is to decompose the high-dimensional distribution into lower-dimensional ones. Given an

instance $M$, let $x_d$ denote the $d^{th}$ element of a solution vector $x$. In Nair et al. (2020), it is assumed that the variables are independent of each other, i.e.,

$$P_\theta(x; M) = \prod_{d=1}^{n} p_\theta(x_d; M). \tag{5}$$

With this assumption, the high-dimensional sampling problem is decomposed into $n$ 1-dimensional sampling problems for each $x_d$ according to their probabilities $p_\theta(x_d; M)$. Since $p_\theta(x_d = 1; M) = 1 - p_\theta(x_d = 0; M)$, we only need $p_\theta(x_d = 1; M)$ for $d \in \{1, 2, ..., n\}$ to represent the conditional probability $P_\theta(x; M)$. Then the conditional distribution mapping outputs a $n-$dimension vector $(p_\theta(x_1 = 1; M), ..., p_\theta(x_n = 1; M))$. Hence, the prediction of the GNN model can be represented as $F_\theta(M) \equiv (\hat{p}_1, \hat{p}_2, ..., \hat{p}_n)$, where $\hat{p}_d \equiv p_\theta(x_d = 1; M)$ for $d \in \{1, 2, ..., n\}$.

Let $S_d^i \subseteq \{1, 2, ..., N^i\}$ denote the set of indices in $L^i$ with their $d^{th}$ component being 1.

$$p_d^i \equiv \sum_{j \in S_d^i} w^{i,j}, \tag{6}$$

where $p_d^i$ is normalized by $|L_i|$. Given an MILP instance $M$, we can calculate a corresponding learning target in the form of vector, i.e, $P \equiv (p_1, p_2, ..., p_q)$, where each component is obtained by applying Equation (6). This equation calculates the marginal probability, where the weight $w^{i,j}$ is 1 if the variable holds a value of 1 in the corresponding solution and 0 otherwise. We define such a learning target in the form of a vector as *marginal probabilities*. Solutions of higher quality will be associated with larger weighting coefficients $w^{ij}$ and hence contribute more to the loss function.

For the loss function shown in Equation (4), based on the assumption in Equation (5) and the calculation of probabilities in Equation (4), we have:

$$
\begin{aligned}
L(\theta) &= -\sum_{i=1}^{N} \sum_{d=1}^{n} \sum_{j=1}^{N_i} w^{i,j} \log p_\theta\left(x_d^{i,j}; M^i\right) \\
&= -\sum_{i=1}^{N} \sum_{d=1}^{n} \left\{ \sum_{j \in S_d^i} w^{i,j} \log p_\theta\left(x_d^{i,j}; M^i\right) + \sum_{j \notin S_d^i} w^{i,j} \log p_\theta\left(x_d^{i,j}; M^i\right) \right\} \\
&= -\sum_{i=1}^{N} \sum_{d=1}^{n} \left\{ p_d^i \log\left(\hat{p}_d^i\right) + \left(1 - p_d^i\right) \log\left(1 - \hat{p}_d^i\right) \right\}.
\end{aligned}
$$

This indicates that the multi-dimensional distribution learning loss $L(\theta)$ becomes a summation of each component's probability learning loss. Thus, with Equation (5), the distribution learning is converted to a marginal probabilities learning.

## 3.2 SEARCH

With marginal probabilities as inputs, we adopted a trust region like method to carry out a search algorithm. In this section, we first introduce our observation that the distance between the solution obtained from a rounded learning target and the optimal solution can be very small. Hence, we adopt a trust region like method to address aforementioned challenge (**II**) and present a proposition to manifest the superiority of our framework. The complete framework is shown in Algorithm 1.

### 3.2.1 OBSERVATION

A variable's marginal probability is closer to 1 if this variable takes a value of 1 in the optimal solution, and 0 otherwise. Given an MILP instance $M$ and its learning target $P$, we set the partial solution size parameter $(k_0, k_1)$ to represent the numbers of 0's and 1's in a partial solution. Let $I_0$ denote the set of indices of the $k_0$ smallest components of $P$, and $I_1$ denote the set of indices of the $k_1$ largest components of $P$. If $d \in I \equiv I_1 \cup I_0$, we get a partial solution $\overline{x}_I$ by:

$$\overline{x}_d \equiv \begin{cases} 0 & \text{if } d \in I_0, \\ 1 & \text{if } d \in I_1. \end{cases} \tag{7}$$

Let $x^*$ denote an optimal solution of $M$, empirically, we found $\overline{x}_I$ is close to $x_I^*$ as discussed in Appendix D. Explicitly, we measure the distance by $\ell_1$ norm, and there still exists a small $\Delta > 0$, such that $\|\overline{x}_I - x_I^*\|_1 < \Delta$ while $(k_0 + k_1)$ is a large number. We speculate that, since the optimal solution has the largest weight as shown in equation (6), it is closer to the learning target than all other solutions. As a result, we hypothesize that similar phenomena can be observed with a slightly larger $\Delta$ and the same $(k_0,\ k_1)$ when obtaining the set of indices $I$ based on prediction $F_\theta(M)$.

With this observation, it is reasonable to accelerate the solving process for MILP problems by fixing variables in the partial solution. Specifically, the sub-problem of an instance $M$ using the fixing strategy with the partial solution $\overline{x}_I$ can be formulated as:

$$\min_{x \in D \cap S(\overline{x}_I)} c^\top x, \tag{8}$$

where $S(\overline{x}_I) \equiv \{x \in \mathbb{R}^n : x_I = \overline{x}_I\}$. However, once $\overline{x}_I \neq x_I^*$, the fixing strategy may lead to suboptimal solutions or even infeasible sub-problems, which can also be observed in Appendix C.

### 3.2.2 SEARCH WITHIN A NEIGHBORHOOD

Based on the observations and analysis above, we design a more practicable method. Inspired by the trust region method, we use the partial solution as a starting point to establish a trust region and search for a trial step. The trial step is then applied to the starting point to generate a new solution.

Specifically, given instance $M$, we acquire the set $I$ via the prediction $F_\theta(M)$ from a trained GNN, and get a partial solution $\hat{x}_I$ by equation (7) to construct the trust region problem for a trial step $d^*$ similar to problem (1), (2). At last, output the point updated by trial step. Such a trust region problem is equivalent to following:

$$\min_{x \in D \cap B(\hat{x}_I, \Delta)} c^\top x, \tag{9}$$

where $B(\hat{x}, \Delta) \equiv \{x \in \mathbb{R}^n : \|\hat{x}_I - x_I\|_1 \leq \Delta\}$ denotes a neighborhood constraint. In order to reduce computational costs, we solve this problem only once to find a near-optimal solution. We show that our proposed method always outperforms fixing-based methods in Proposition 1.

**Proposition 1.** *let $z^{Fixing}$ and $z^{Search}$ denote optimal values to problems (8) and (9) respectively. $z^{Search} \leq z^{Fixing}$, if they have same partial solution $\hat{x}_I$.*

**Proof.** *Note that $S(\hat{x}_I) = B(\hat{x}_I, 0) \subset B(\hat{x}_I, \Delta)$, it is obvious*

$$\min_{x \in D \cap B(\hat{x}_I, \Delta)} c^\top x \leq \min_{x \in D \cap S(\hat{x}_I)} c^\top x,$$

*i.e. $z^{Search} \leq z^{Fixing}$.*

$\square$

This proposition demonstrates the advantage of our proposed search strategy over fixing strategy; that is, when fixing strategies may lead to suboptimal solutions or even infeasible sub-problems as a consequence of inappropriate fixing, applying such a trust region search approach can always add flexibility to the sub-problem. Computational studies also provide empirical evidence in support of this proposition. Algorithm 1 presents the details of our proposed search algorithm. It takes the prediction $F_\theta(M)$ as input, and acquires a partial solution $x_d$ and neighborhood constraints. A complete solution $x$ is then attained by solving the modified instance $M'$ with neighborhood constraints appended. The solving process is denoted by $SOLVE(M')$, i.e. utilizing an MILP solver to address instance $M'$.

## 4 COMPUTATIONAL STUDIES

We conducted extensive experiments on four public datasets with fixed testing environments to ensure fair comparisons.

**Benchmark Problems** Four MILP benchmark datasets are considered in our computational studies. Two of them come from the NeurIPS ML4CO 2021 competition Gasse et al. (2022), including the *Balanced Item Placement* (denoted by IP) dataset and the *Workload Appointment* (denoted by WA) dataset (Gasse et al., 2022). We generated the remaining two datasets: *Independent Set* (IS) and

---

**Algorithm 1** Predict-and-search Algorithm

---

**Parameter**: Size $\{k_0, k_1\}$, radius of the neighborhood: $\Delta$
**Input**: Instance $M$, Probability prediction $F_\theta(M)$
**Output**: Solution $x \in \mathbb{R}^n$

1:  Sort the components in $F_\theta(M)$ from smallest to largest to obtain sets $I_0$ and $I_1$.
2:  **for** $d = 1 : n$ **do**
3:      **if** $d \in I_0 \cup I_1$ **then**
4:          **create binary variable** $\delta_d$
5:          **if** $d \in I_0$ **then**
6:              **create constraint**
                    $x_d \le \delta_d$
7:          **else**
8:              **create constraint**
                    $1 - x_d \le \delta_d$
9:          **end if**
10:     **end if**
11: **end for**
12: **create constraint** $\sum\limits_{d \in I_0 \cup I_1} \delta_d \le \Delta$
13: Let $M'$ denote the instance $M$ with new constraints and variables
14: Let $x = SOLVE(M')$
15: **return** $x$

---

*Combinatorial Auction* (CA) using Ecole library (Prouvost et al., 2020) as Gasse et al. (2019) did. Note that the other two datasets from Gasse et al. (2019), *Set Covering* and *Capacitated Facility Location*, are not chosen since they can be easily solved by state-of-the-art MILP solvers, such as Gurobi and SCIP. Details of our selected datasets can be found in the Appendix F.

**Graph neural networks** A single layer perceptron embeds feature vectors so they have the same dimension of 64, and the layer normalization Ba et al. (2016) is applied for a better performance of the network. Then we adopted 2 half-convolution layers from Gasse et al. (2019) to conduct information aggregation between nodes.

Finally, marginal probabilities are obtained by feeding the aggregated variable nodes into a 2-layer perceptron followed by a sigmoid activation function.

**Training protocol** Each dataset contains 400 instances, including 240 instances in the training set, 60 instances in the validation set, and 100 instances in the test set. All numerical results are reported for the test set. Our model is constructed with the PyTorch framework, and the training process runs on GPUs. The loss function is specified in Equation (4) and (5) with a batch size of 8. The ADAM optimizer Kingma & Ba (2014) is used for optimizing the loss function with a start learning rate of 0.003.

**Evaluation metrics** For each instance, we run an algorithm of interest and report its incumbent solution's objective value as OBJ. Then we run single-thread Gurobi for $3,600$ seconds and denote as BKS the objective to the incumbent solution returned. BKS is updated if it is worse than OBJ. We define the absolute and relative primal gaps as: $\mathrm{gap}_{\mathrm{abs}} \equiv |\mathrm{OBJ} - \mathrm{BKS}|$ and $\mathrm{gap}_{\mathrm{rel}} \equiv |\mathrm{OBJ} - \mathrm{BKS}| / (|\mathrm{BKS}| + 10^{-10})$, respectively and utilize them as performance metrics. Clearly, a smaller primal gap indicates a stronger performance.

**Evaluation Configurations** All evaluations are performed under the same configuration. The evaluation machine has two Intel(R) Xeon(R) Gold 5117 CPUs @ 2.00GHz, 256GB ram and two Nvidia V100 GPUs. SCIP 8.0.1 Bestuzheva et al. (2021), Gurobi 9.5.2 Gurobi Optimization, LLC (2022) and PyTorch 1.10.2 Paszke et al. (2019) are utilized in our experiments. The emphasis for Gurobi and SCIP is set to focus on finding better primal solutions. The time limit for running each experiment is set to $1,000$ seconds since a tail-off of solution qualities was often observed after that.

**Data collection** Details will be provided in the Appendix B.

## 5 RESULTS AND DISCUSSION

To investigate the benefits of applying our proposed predict-and-search framework, we conducted comprehensive computational studies that: (i) compare our framework with SCIP and Gurobi; (ii) compare our framework with Neural Diving framework (Nair et al., 2020). Other numerical experiments including comparing against a modified version of confidence threshold neural diving (Yoon, 2022) along with implementation details can be found in the Appendix D.

### 5.1 COMPARING AGAINST STATE-OF-THE-ART SOLVERS

In what follows, we utilize SCIP (Gurobi) as an MILP solver in our proposed framework and denote our approach as PS+SCIP (PS+Gurobi). Figure 2 exhibits the progress of average $\text{gap}_{\text{rel}}$ as the solving process proceeds. In Figure 2a, we notice a significant performance improvement of our framework (blue) upon default SCIP (green), and such an improvement is also observed for WA, CA and IS datasets as shown in 2b, 2c and 2d. Compared with default Gurobi (black), our approach(red) still performs better, specially in the IS dataset. We remark that, in Figure 2d, PS+Gurobi obtained optimal solutions to IS problems within 10 seconds. The performance comparison in Figure 2 also indicates that our proposed framework can bring noticable performance improvement upon MILP solvers, regardless of the solver choice.

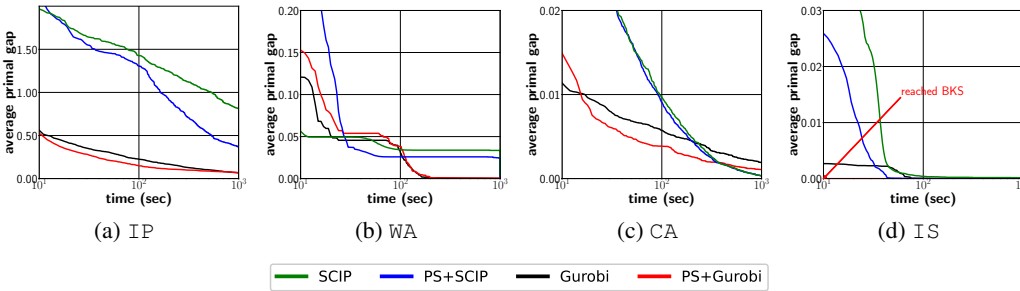

Figure 2: Performance comparisons between PS, Gurobi and SCIP, where the y-axis is relative primal gap averaged across 100 instances; each plot represents one benchmark dataset.

In Table 1, we present objective values averaged across 100 instances at the time limit of $1,000$ seconds. Column "$\text{gap}_{\text{abs}}$" provides the absolute primal gap at $1,000$ seconds, and column "gain" presents the improvement of our method compared over an MILP solver. For instance, Table 1 shows that BKS for IP dataset is 12.02, and the absolute primal gaps of SCIP and PS+SCIP are 7.41 and 3.44 respectively. Then gain is computed as $(7.41 - 3.44)/7.41 \times 100\% = 53.6\%$. According to Table 1, we can claim that our proposed framework outperforms SCIP and Gurobi with average improvements of $51.1\%$ and $9.9\%$, respectively. We notice that the improvement for Gurobi is not as significant as for SCIP since Gurobi is significantly more capable in terms of identifying high-quality solutions than SCIP.

Table 1: Average objective values given by different approaches at 1,000 seconds.

| dataset | BKS | SCIP | | PS+SCIP | | gain | Gurobi | | PS+Gurobi | | gain |
|---|---|---|---|---|---|---|---|---|---|---|---|
| | | OBJ | $\text{gap}_{\text{abs}}$ | OBJ | $\text{gap}_{\text{abs}}$ | | OBJ | $\text{gap}_{\text{abs}}$ | OBJ | $\text{gap}_{\text{abs}}$ | |
| IP | 12.02 | 19.43 | 7.41 | 15.46 | **3.44** | **53.6%** | 12.65 | **0.63** | 12.71 | 0.69 | -9.5% |
| WA | 700.94 | 704.23 | 3.29 | 702.35 | **1.41** | **57.1%** | 701.24 | 0.30 | 701.22 | **0.28** | **6.7%** |
| IS | 685.04 | 684.94 | 0.10 | 685.03 | **0.01** | **90.0%** | 685.04 | 0.00 | 685.04 | 0.00 | 0.00 |
| CA | 23,680.01 | 23,671.66 | 8.35 | 23,671.95 | **8.06** | **3.5%** | 23,635.07 | 44.94 | 23,654.47 | **25.54** | **43.2%** |
| avg. | | | | | | **51.1%** | | | | | **9.9%** |

## 5.2 COMPARING AGAINST NEURAL DIVING

Another interesting comparison should be conducted against the state-of-the art ML method for optimization: the Neural Diving framework with Selective Net. However, since detailed settings and codes of the original work Nair et al. (2020) are not publicly available, reproducing the exact same results is impractical. To our best effort, a training protocol with fine parameter tuning and an evaluation process are established following algorithms provided in Nair et al. (2020). Three of six tested benchmark datasets used in Nair et al. (2020) are publicly available: *Corlat*, *Neural Network Verification*, and *MipLib*. Most Corlat instances can be solved by SCIP within a few seconds; MipLib contains instances with integer variables rather than binary variables, which is out of the scope of this work. Hence, Neural Network Verification (denoted as NNV) is chosen as the benchmark dataset for the comparison study. It is noteworthy that, empirically, turning on the presolve option in SCIP (Bestuzheva et al., 2021) causes false assertion of feasibility on many NNV instances. Hence, in our experiments on the NNV dataset, the presolve option is turned off, which potentially hurts the performances of both SCIP itself and frameworks implemented with SCIP.

Under such circumstances, the best performance obtained is exhibited in Figure 3a. Clearly, the Neural Diving framework achieves significant improvement over default SCIP. With such an imple-

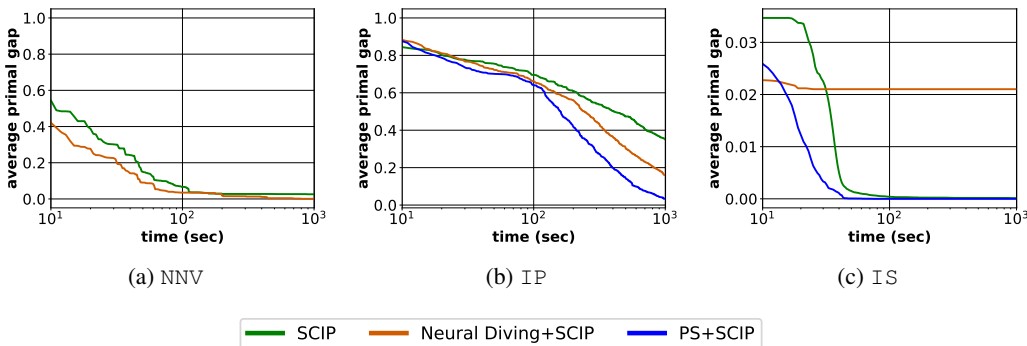

Figure 3: The experiment shown in Figure 3a validates the performance of our Neural Diving implementation by comparing it with default SCIP on NNV dataset. Figure 3b and 3c exhibit performance comparisons between PS and Neural Diving framework on IP and IS datasets. All methods related are implemented with SCIP. The result shows that our proposed method outperforms the Neural Diving framework significantly.

mentation, we can start to compare our proposed framework against the Neural Diving approach. Due to the limitation of computational power, we are not able to find suitable settings of parameters to train Neural Diving framework on WA and CA datasets. Hence we conducted experiments only on IP and IS datasets. As shown in Figure 3b and 3c, our predict-and-search framework produced at least three times smaller average relative primal gaps. An interesting observation is that Neural Diving framework failed to surpass SCIP on IS dataset where the optimality is hard to achieve, while our framework outperformed both SCIP and the implemented Neural Diving method.

## 6 CONCLUSIONS

We propose a predict-and-search framework for tackling difficult MILP problems that are routinely solved. A GNN model was trained under a supervised learning setting to map from bipartite graph representations of MILP problems to marginal probabilities. We then design a trust region based algorithm to search for high-quality feasible solutions with the guidance of such a mapping. Both theoretical and empirical supports are provided to illustrate the superiority of this framework over fixing-based strategies. With extensive computational studies on publicly available MILP datasets, we demonstrate the effectiveness of our proposed framework in quickly identifying high-quality feasible solutions. Overall, our proposed framework achieved 51.1% and 9.9% better primal gaps comparing to SCIP and Gurobi, respectively.

## ACKNOWLEDGMENTS

This work is supported by the National Key R&D Program of China under grant 2022YFA1003900; Huawei; Hetao Shenzhen-Hong Kong Science and Technology Innovation Cooperation Zone Project (No.HZQSWS-KCCYB-2022046); University Development Fund UDF01001491 from The Chinese University of Hong Kong, Shenzhen; Guangdong Key Lab on the Mathematical Foundation of Artificial Intelligence, Department of Science and Technology of Guangdong Province.

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

## A  HALF CONVOLUTION

We use two interleaved half-convolutions Gasse et al. (2019) to replace the graph convolution. After that, the GNN we used can be formulated as:

$$h_{v_i}^{(k)} = \text{MLP}_c^{(k)} \left( h_{v_i}^{(k-1)}, \sum_{u \in N(v_i)} h_u^{(k-1)} \right) \quad i \in \{n, n+1, ..., n+m-1, n+m\},$$

$$h_{v_i}^{(k)} = \text{MLP}_v^{(k)} \left( h_{v_i}^{(k-1)}, \sum_{u \in N(v_i)} h_u^{(k-1)} \right) \quad i \in \{1, 2, ..., n\},$$

where $\text{MLP}_c^{(k)}$, $\text{MLP}_v^{(k)}$ and $g(\cdot)$ are 2-layer perceptrons with ReLU activation functions. That is, a half-convolution is performed to promote each constraint node aggregating information from its relevant variable nodes; after that, another one is performed on each variable node to aggregate information from its relevant constraint nodes and variable nodes.

## B  DATA COLLECTION

The training process requires bipartite graph representations of MILP problems as the input and marginal probabilities of variables as the label. We first extract the bipartite graph by embedding variables and constraints as respective feature vectors (see Appendix E). Then, we run a single-thread Gurobi with a time limit of 3,600 seconds on training sets to collect feasible solutions along with their objective values. These solutions are thereafter weighted via energy functions as in Equation (3) to obtain marginal probabilities.

## C  COMPARING OBJECTIVE VALUES IN DIFFERENT PARTIAL SOLUTIONS

We also scrutinize why fixing strategy could fail. Variables in the optimal solution are randomly perturbed to simulate a real-world setting that a prediction is very likely to be inaccurate. Figure 4 exhibited a trend that, as we perturb more variables, the absolute primal gap (green) increases, and the percentage of infeasible sub-problems (red) increases. The absolute gaps shown in Figure 4a indicate large performance drawbacks given that the optimal objective value is 685. Convincingly, we conclude that fixing approaches presumably produce sub-optimal solutions. Besides, as shown in Figure 4b, randomly perturbing one variable can result in 20% of infeasible sub-problems. That is, fixing strategy could lead to infeasible sub-problems even if relatively accurate predictions are provided.

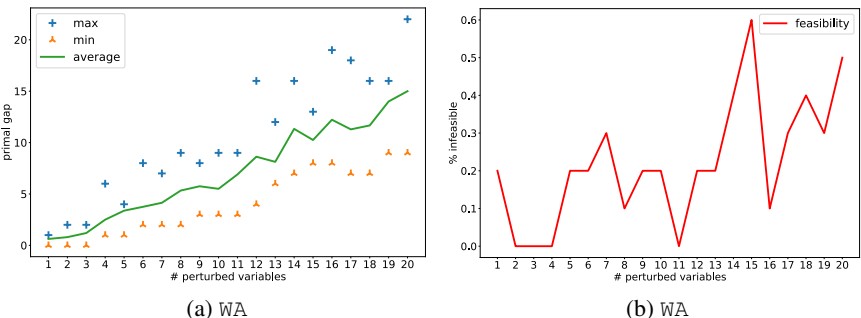

(a) WA                    (b) WA

Figure 4: This plot shows how qualities of solutions to sub-problems vary as variables are randomly perturbed in one WA instance. Maximum, minimum and average values are presented in the plot. The x-axis is the number of variables perturbed in the partial solution while y-axis of Figure 4a is the absolute primal gap between average objective values (only include feasible sub-problems) and the optimal objective value; y-axis of Figure 4b on the right is the percentage of infeasible sub-problems.

## D    COMPARING AGAINST A MODIFIED VERSION OF CONFIDENCE THRESHOLD NEURAL DIVING

In this part, we demonstrate our advantage over a fixing strategy by conducting numerical experiments. In Yoon (2022)'s work, thresholds are chosen to determine how many variables to fix. We modify this method so that the number of variables becomes a tunable hyper-parameter, which makes the procedure more controllable and flexible. We directly use the learning target to define confidence scores, and we obtain partial solutions by setting different size parameters $(k_0, k_1)$ as stated in Section 3.2.1. $\ell_1$ norm is utilized for measuring such distances. We selected $(700, 0)$, $(750, 0)$, $(800, 0)$, $(850, 0)$, and $(900, 0)$ as size parameters and obtained their corresponding $\ell_1$ distances as $0.00$, $0.04$, $1.20$, $35.26$, and $85.01$. We observed that such distances can be small if only a small portion of variables are taken into consideration. However, as we enlarge the number of involved variables, the distance increases dramatically. Hence, the predict-and-search approach can involve larger set of variables than fixing strategy does while still retaining the optimality.

To support this argument, we compare the performance of our approach with that of a fixing strategy. For simplicity, we denote the modified version of confidence threshold neural diving as Fixing in Figures. Figure 5 shows the average relative primal gap achieved by different approaches, where both approaches use a fixed number of selected variables. One can spot the dominance of our approach over the modified version of confidence threshold neural diving from Figure 5a; this implies that such a fixing strategy only leads to mediocre solutions, while the search method (red) achieves solutions with better qualities. In Figure 5d, both approaches are capable of finding optimal solutions instantly, but the ones obtained by fixing method is far from optimal solutions to original problems. We notice that in Figure 5b and 5c, our framework struggles to find good solution in the early solving phases, but produces equivalent or better solutions within $1,000$ seconds comparing to the fixing method. A possible reason is that our solution results in larger feasible regions comparing to the fixing strategy. Conclusively, our framework always outperforms the fixing strategy.

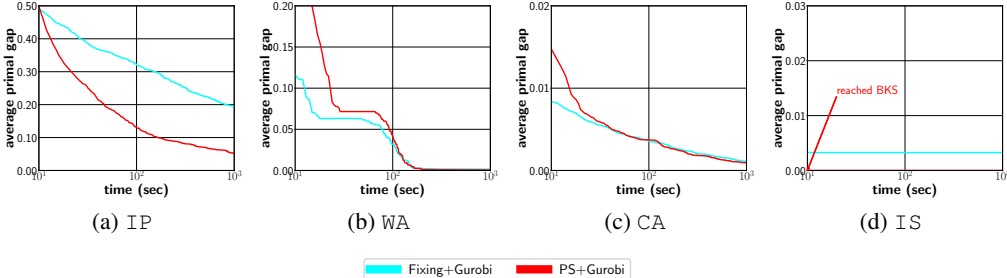

Figure 5: This figure shows average $\mathrm{gap}_{\mathrm{rel}}$ achieved by the search method and the fixing method under the same partial solution. The results are averaged across 100 instances, and each plot represents one dataset. The proposed framework shows a constantly dominant performance over the fixing-based method.

## E    FEATURE DESCRIPTIONS FOR VARIABLE NODES, CONSTRAINT NODES AND EDGES

To encode MILP problems, we propose a set of features extracted from constraints, variables, and edges. This set of features is relatively light-weighted and generalized; each feature is obtained either directly from the original MILP model or by conducting simple calculations. Moreover, such extractions do not require pre-solving the MILP instance, which would save a significant amount of time for large and difficult MILP problems.

Table 2: Features in embedded bipartite representations

|  | # features. | name | description |
|---|---|---|---|
| | 1 | obj | normalized coefficient of variables in the objective function |
| | 1 | v_coeff | average coefficient of the variable in all constraints |
| | 1 | Nv_coeff | degree of variable node in the bipartite representation |
| | 1 | max_coeff | maximum value among all coefficients of the variable |
| Variable | 1 | min_coeff | minimum value among all coefficients of the variable |
| | 1 | int | binary representation to show if the variable is an integer variable |
| | 12 | pos_emb | binary encoding of the order of appearance for each variable among all variables. |
| | 1 | c_coeff | average of all coefficients in the constraint |
| Constraint | 1 | Nc_coeff | degree of constraint nodes in the bipartite representation |
| | 1 | rhs | right-hand-side value of the constraint |
| | 1 | sense | the sense of the constraint |
| Edge | 1 | coeff | coefficient of variables in constraints |

## F  SIZES OF BENCHMARK PROBLEMS

Table 3 exhibits dimensions of the largest instance of each tested benchmark dataset. The numbers of constraints, variables, and binaries are presented.

Table 3: Maximum problem sizes of each dataset

| dataset | # constr. | # var. | # binary var. |
|---|---|---|---|
| IP | 195 | 1,083 | 1,050 |
| WA | 64,480 | 61,000 | 1,000 |
| IS | 600 | 1,500 | 1,500 |
| CA | 6,396 | 1,500 | 1,500 |

## G  PARAMETRIC SETTINGS FOR EXPERIMENTS

For experiments where our predict-and-search framework is compared with SCIP, Gurobi, and fixing-based strategy, the settings for fixing-parameter $(k_0, k_1)$ and the neighborhood parameter $\Delta$ are listed in Table 4. Based on the performance of the underlying solver, various settings of $(k_0, k_1)$ are used to carry out experiments for each benchmark dataset shown in Table 4. The radius of the search area $\Delta$ is chosen respectively for different implementations (PS+SCIP, PS+Gurobi, and Fixing+SCIP) of our framework as shown in Table 4.

Table 4: $k_0$, $k_1$, and $\Delta$ settings for different dataset

| dataset | PS+SCIP | | PS+Gurobi | | Fixing+SCIP | |
|---|---|---|---|---|---|---|
| | $k_0, k_1$ | $\Delta$ | $k_0, k_1$ | $\Delta$ | $k_0, k_1$ | $\Delta$ |
| IP | 400, 5 | 1 | 400, 5 | 10 | 400, 5 | 0 |
| WA | 0, 500 | 5 | 0, 500 | 10 | 0, 500 | 0 |
| IS | 300, 300 | 15 | 300, 300 | 20 | 300, 300 | 0 |
| CA | 400, 0 | 10 | 600, 0 | 1 | 600, 0 | 0 |

