# OpenReview forum: "A GNN-Guided Predict-and-Search Framework for Mixed-Integer Linear Programming"
_ICLR.cc/2023/Conference — ICLR 2023 poster_

### Official Review · Reviewer_E5hv · 2022-10-19

**Confidence:** 4
**Correctness:** 2
**Technical Novelty And Significance:** 2
**Empirical Novelty And Significance:** 2
**Recommendation:** 3

**Clarity, Quality, Novelty And Reproducibility:**

Clarity: The basic idea of the paper is easily understandable. Authors do a reasonable job describing the intuition behind their work. However, authors make a few mistakes in mathematical notations and description of experiments, which poses a moderate difficulty clearly understanding the paper.

Quality/Novelty: The proposed method is a relatively simple refinement of Neural Diving. However, the strategy of using L1-constrained optimization instead of fixing variables seems like a practically useful & broadly applicable idea. Hence this strategy might indeed make a good impact if its empirical advantage against Neural Diving is well-established. As mentioned in Strengths & Weaknesses, however, authors don't compare against full-fledged Neural Diving with SelectiveNet, and thus it is a bit difficult to evaluate whether the proposed method actually has an empirical advantage.

Reproducibility: I don't see source code, generated datasets, or scripts being shared in supplementary material. Hence reproducing this paper will take a considerable effort re-implementing the training code as well as integration with SCIP/Gurobi.

A few detailed notes on clarity:

Section 3.1.2. Equation (7). It's worth mentioning this factorization is also proposed from Nair et al.

In equation (8), shouldn't this be normalized by $|L_i|$ to be a value between 0 and 1?

Under equation 11, $||\hat{x}(I) - \hat{x}(I)||$ seems to be a typo of $||\hat{x}(I) - x(I)||$.

As authors themselves mention, proposition 1 is quite obvious. Usually such obvious fact should just be mentioned in the narrative rather than dedicating a proposition.

gap_p in Table 1 doesn't seem to be explained. It seems like the percentage of the improvement of the gap.

In Section 5.1, I wonder how $(k_0, k_1)$ and $\Delta$ were chosen in these experiments?

In Section 5.2, $(k_0, k_1)$ values are repeated as (700, 0) four times. Seems like a mistake?

**Details Of Ethics Concerns:**

Using ML models for primal heuristics might introduce some bias in MIP solutions, but the bias/fairness concerns of MIP problems are to my knowledge not very well known, so it is difficult for me to assess the consequence of the introduced bias.

**Strength And Weaknesses:**

The main strengths are the simplicity of the approach and the practical appeal. It is conceivable predicted variables from ML model is often not feasible, hence making only a "soft" commitment and searching in an L1 ball seems like a practically useful solution. The method is also conceptually simple and intuitive.

The main weakness of this paper is that the paper builds upon Neural Diving of Nair et al (2021) but authors don't compare against Neural Diving. While Neural Diving uses SelectiveNet to identify variables to fix, authors simply rely on confidence scores. Hence it is unclear whether the proposed predict-and-search (PS) method is superior to fixing strategy, or this is just the consequence of using a naive method for uncertainty quantification. I understand an open-source implementation of Neural Diving is not available (as far as I know), but given high similarity of the proposed method and Neural Diving, I still think authors should've implemented SelectiveNet (the only component that seems to be missing in the current implementation) and compare against full Neural Diving algorithm from Nair et al.

**Summary Of The Paper:**

Authors propose a primal heuristic for solving MIP, following much of Neural Diving strategy from Nair et al (2021). Instead of fixing predicted variables from ML model, authors propose to solve a modified MIP problem which constrains the solution space to be the L1 ball around the predicted solution. Authors demonstrate across two datasets from ML4CO competition and two synthetic datasets from Ecole package that the proposed method works better than the baseline of fixing variables from predicted solution.

**Summary Of The Review:**

While the paper heavily builds upon Neural Diving of Nair et al (2021), authors deviate from Neural Diving in their decision to not use SelectiveNet for uncertainty quantification. Hence, it is unclear whether the claimed improvement over the fixing strategy is the consequence of using a naive uncertainty quantification method, or the benefit of the proposed Predict-and-Search strategy.

---

> ### Author Response · Authors · 2022-11-19
> **Official Response to Reviewer E5hv**
>
> We thank reviewer E5hv so much for the careful review and insightful comments!  It seems the reviewer's biggest concern is whether our method is a simple extension of the Neural Diving method[1]. As suggested, we have updated our paper based on the comments of the reviewers. We hope our following point-to-point response can address your concerns and clarify our contribution. Summary of revision:
> - Thanks for your suggestion. We have added computational studies to compare our proposed approach against the full-fledged Neural Diving with SelectiveNet [1]. Please check the Appendix section IV. First, we implemented the Neural Diving method that could reproduce a similar performance on the NNV dataset as shown in [1]. Then, we tested it on datasets mentioned in our paper (with fine-tuning). The reported numerical results indicate that our proposed Predict-and-Search method indeed performs better.
> - For reproducibility, we have now released the code of the proposed approach at https://anonymous.4open.science/r/predict-and-search-0F6F.
> - We have fixed some incorrect expressions and typos.
>
> > **Q1**: Section 3.1.2. Equation (7). It's worth mentioning this factorization is also proposed from Nair et al.
>
> Thanks for pointing this out. Now we have clarified this in our revised submission.
>
> > **Q2**: In equation (8), shouldn't this be normalized by $|L_i|$ to be a value between 0 and 1?
>
> Yes, thanks for raising this. Now we have clarified this in the revised submission.
>
> > **Q3**:  Under equation 11, $||\hat {x}(I)−\hat{x}(I)||$ seems to be a typo of $||\hat {x}(I)−{x}(I)||$
> gap_p in Table 1 doesn't seem to be explained. It seems like the percentage of the improvement of the gap.
> In Section 5.2, $(k_0, k_1)$ values are repeated as (700, 0) four times. Seems like a mistake?
>
> Thanks for the careful review! We have fixed it in the new submission.
>
> > **Q4**: As authors themselves mention, proposition 1 is quite obvious. Usually such obvious fact should just be mentioned in the narrative rather than dedicating a proposition.
>
> Thanks for raising this point. Though this proposition seems obvious, it would help us differentiate our predict-and-search framework from the works of [1,2] and hence improve our paper's readability.
>
> > **Q5**: In Section 5.1, I wonder how $(k_0, k_1)$ and $\Delta$ were chosen in these experiments?
>
> Regarding the selection of three hyperparameters, we first set $\Delta$ to 0. Then, We tune $(k0, k1)$ by setting them to involve various percentages of decision variables, such as 30%, 40%, and 50%. After acquiring a proper setting on $(k_0, k_1)$, we set $\Delta$ to 1, 5, 15, etc., and evaluate the performance of our Predict-and-Search; the $\Delta$ leading to the best objective value will be selected, and hence we finalize the settings of  $(k_0, k_1, \Delta)$.
>
> For clarity, we have put the algorithm into the main body of the paper.
>
> [1] Vinod Nair, Sergey Bartunov, Felix Gimeno, Ingrid von Glehn, Pawel Lichocki, Ivan Lobov, Brendan O’Donoghue, Nicolas Sonnerat, Christian Tjandraatmadja, Pengming Wang, Ravichandra Addanki, Tharindi Hapuarachchi, Thomas Keck, James Keeling, Pushmeet Kohli, Ira Ktena, Yujia Li, Oriol Vinyals, and Yori Zwols. Solving mixed integer programs using neural networks, 2020.

---

### Official Review · Reviewer_BsbB · 2022-10-24

**Confidence:** 3
**Correctness:** 4
**Technical Novelty And Significance:** 3
**Empirical Novelty And Significance:** 2
**Recommendation:** 6

**Clarity, Quality, Novelty And Reproducibility:**

What is the "pre-processing" time required by the new algorithm: e.g., model training, but also the time required to select hyperparameters like Delta, k_0, and k_1?

**Strength And Weaknesses:**

The proposed method is a natural and reasonable upon ML-based heuristics that are based on variable fixing. While this extension is not particularly deep and uses "off-the-shelf" concepts, the computational results seem to indicate its utility. The paper is extremely well-written and does a great job introducing each of the concepts used to develop the algorithm.

Why did the authors not compare against the ML-based fixing algorithms that they reference in the introduction? That seems to be the fairest baseline for comparison, as these algorithms are the most similar in setting and approach to the one presented by the authors.

I would appreciate more detail (in the appendix, but also some in the text) about the implementation details of the algorithm. For example, I am curious how the algorithm is implemented with SCIP and Gurobi -- presumably it is used via a heuristic callback with both solvers? If so, can the authors break out how much time is spent in the callback (and how many calls), and what fraction of the total allotted wall time this eats up?



**Summary Of The Paper:**

This paper presents a heuristic for constructing good feasible solutions for families of related mixed-integer linear programming (MILP) problems. The heuristic proceeds by taking an instance from a given MILP family and 1) using a machine learning model to output weights (roughly: approximate solution values) for each decision variable, then 2) solve a sub-MILP in the (small) neighborhood around the point proposed by the predictive model. Computational results suggest that this heuristic can outperform the suite of heuristics in SCIP and Gurobi.

**Summary Of The Review:**

The paper is very well-written, and provides a natural but worthwhile extension of a recent stream of algorithms for producing heuristic solutions to families of related MILP instances. My concerns are mostly based around the baseline for comparison used in the computational section.

---

> ### Author Response · Authors · 2022-11-19
> **Official Response to Reviewer BsbB**
>
> We sincerely thank Reviewer BsbB for their positive feedback and precious suggestions! Below we address every comment in detail:
>
> > **Q1**: Why did the authors not compare against the ML-based fixing algorithms that they reference in the introduction? That seems to be the fairest baseline for comparison, as these algorithms are the most similar in setting and approach to the one presented by the authors.
>
> Thank you for your suggestions. We have added computational studies to compare our proposed approach against the Neural Diving method[1]. Please check Appendix in section IV.
>
> > **Q2**: I would appreciate more detail (in the appendix, but also some in the text) about the implementation details of the algorithm. For example, I am curious how the algorithm is implemented with SCIP and Gurobi -- presumably it is used via a heuristic callback with both solvers? If so, can the authors break out how much time is spent in the callback (and how many calls), and what fraction of the total allotted wall time this eats up?
>
> Thanks for your advice. We do not use callbacks. Specifically, for a given MILP problem, our approach is to create region constraints using the output of the neural network as the initial point and add them to the original problem. Finally, we let SCIP/Gurobi to solve the “tightened” MILP problem. Our code is available at https://anonymous.4open.science/r/predict-and-search-0F6FIn . In this repository, the implementation details of our algorithm can be found (such as implemented with SCIP /Gurobi, model training, etc.).
>
> > **Q3**: What is the "pre-processing" time required by the new algorithm: e.g., model training, but also the time required to select hyperparameters like Delta, k_0, and k_1?
>
> Thank you for your suggestion, a detailed discussion of the time consumption of each component will indeed help us analyze experiments. Regarding the “pre-processing” time , we have calculated a table as follows:
> | instance | model training(h) | model loading(s) |   | obtain BG(s) | NN inference(s) |
> |----------|-------------------|------------------|---|--------------|-----------------|
> | IS       | 13                | 0.01             |   | 0.11         | 0.02            |
> | WA       | 76                | 0.01             |   | 2.58         | 0.07            |
> | IP       | 9                 | 0.01             |   | 0.03         | 0.01            |
> | CA       | 6                 | 0.01             |   | 0.06         | 0.01            |
>
> where BG denotes the bipartite graph which is the input of neural network, NN denotes the neural network, and we use GNN[2] in our method.
>
> Regarding the selection of three hyperparameters, we first set Delta to 0. Then, We tune (k0, k1) by setting them to involve various percentages of decision variables, such as 30%, 40%, and 50%. After acquiring a proper setting on (k_0, k_1), we set Delta to 1, 5, 15, etc, and evaluate the performance of our Predict-and-Search framework; the Delta leading to the best objective value will be selected, and then we finalize the settings of  (k_0, k_1, Delta).
>
> [1] Vinod Nair, Sergey Bartunov, Felix Gimeno, Ingrid von Glehn, Pawel Lichocki, Ivan Lobov, Brendan O’Donoghue, Nicolas Sonnerat, Christian Tjandraatmadja, Pengming Wang, Ravichandra Addanki, Tharindi Hapuarachchi, Thomas Keck, James Keeling, Pushmeet Kohli, Ira Ktena, Yujia Li, Oriol Vinyals, and Yori Zwols. Solving mixed integer programs using neural networks, 2020.
>
> [2]Maxime Gasse, Didier Chetelat, Nicola Ferroni, Laurent Charlin, and Andrea Lodi. Exact combi- ´ natorial optimization with graph convolutional neural networks. Advances in Neural Information Processing Systems, 32, 2019.

---

### Official Review · Reviewer_DX2F · 2022-10-25

**Confidence:** 4
**Clarity, Quality, Novelty And Reproducibility:** 1. The presentation is unclear.
2. Th…
**Correctness:** 4
**Technical Novelty And Significance:** 3
**Empirical Novelty And Significance:** 2
**Recommendation:** 5

**Strength And Weaknesses:**

Strengths:
1. The proposed predict-and-search framework is novel.
2. The trust region in the search algorithm is well designed.
3. Experiments demonstrate that the proposed framework improves the performance of SCIP and Gurobi on several datasets.

Weaknesses:
1. The paper is hard to follow as the presentation of the proposed approach is unclear. For example, the definition of the so-called “weighted conditional marginal probability distribution” is confusing.
2. The authors may want to add more baselines to demonstrate the effectiveness of the proposed approach. For example, as the predicting part of the method mainly follows [1], the authors should add [1] as a baseline.
3. The authors may want to further discuss on the connections and differences between the proposed predict-and-search approach and existing predicting based (e.g., [1]) and searching based methods (e.g., [2]).
4. The authors only consider discrete variables to be binary, which limits the applications of the proposed framework.
5. In the 4th line in Section 5.2, why are the values of the size parameters the same?

Minor Issue:
1. The authors may want to use “\citep{}” instead of “\cite{}” for some of the citations.

[1] Vinod Nair, Sergey Bartunov, Felix Gimeno, Ingrid von Glehn, Pawel Lichocki, Ivan Lobov, Brendan O’Donoghue, Nicolas Sonnerat, Christian Tjandraatmadja, Pengming Wang, Ravichandra Addanki, Tharindi Hapuarachchi, Thomas Keck, James Keeling, Pushmeet Kohli, Ira Ktena, Yujia Li, Oriol Vinyals, and Yori Zwols. Solving mixed integer programs using neural networks, 2020.

[2] Li Z, Chen Q, Koltun V. Combinatorial optimization with graph convolutional networks and guided tree search[J]. Advances in neural information processing systems, 2018, 31.

**Summary Of The Paper:**

The authors propose a predict-and-search framework for solving mixed-integer linear programming (MILP) problems. Specifically, they first predict the solution distributions, and then search for near-optimal solutions within a trust region constructed from the prediction. Experiments demonstrate that the proposed framework improves the performances of two state-of-the-art optimization solvers SCIP and Gurobi on several datasets.

**Summary Of The Review:**

Overall, I believe that this paper is marginally below the acceptance threshold. Though the authors propose a novel predict-and-search framework for solving mixed-integer linear programming problem, they miss some important baselines to demonstrate the effectiveness of their proposed framework. Moreover, they may want to improve the writing.

---

> ### Author Response · Authors · 2022-11-19
> **Official Response to Reviewer DX2F**
>
> We sincerely thank Reviewer DX2F for their valuable suggestions. Summary of revision:
>
> - We have adjusted the structure of our paper，and fixed some incorrect expressions and typos;  more references are added as suggested.
>
> - We have added computational studies to compare our proposed approach against the Neural Diving method[1]. Please check Appendix in section IV. The computational results demonstrate that our approach performs better.
>
> Below we address every comment in detail.
>
> > **Q1**: The paper is hard to follow as the presentation of the proposed approach is unclear. For example, the definition of the so-called “weighted conditional marginal probability distribution” is confusing.
>
> Thanks for the suggestion! For the clearness of the presentation, we updated Section 3.1.2 to clarify the concept. The updated version better clarifies our contribution.
>
> > **Q2**: The authors may want to add more baselines to demonstrate the effectiveness of the proposed approach. For example, as the predicting part of the method mainly follows [1], the authors should add [1] as a baseline.
>
> We appreciate the advice. We have added computational studies to compare our proposed approach against the Neural Diving method[1]. Please check the Appendix section IV.
>
> > **Q3**: The authors may want to further discuss on the connections and differences between the proposed predict-and-search approach and existing predicting based (e.g., [1]) and searching based methods (e.g., [2]).
>
> Thank you for bringing this searching based methods to our attention! We appreciate the great efforts made by the author; furthur discussions can be found in the updated paper.
>
> > **Q4**: The authors only consider discrete variables to be binary, which limits the applications of the proposed framework.
>
> Thanks for raising this point.  Indeed, our proposed framework only applies to the case of binary variables. In practice, the vast majority of integer-constrained variables in MILPs are binary variables. In future works, we will extend our approach to general MILP instances.
>
> > **Q5**: In the 4th line in Section 5.2, why are the values of the size parameters the same?
>
> Thanks for pointing out this typo, and we’ve fixed it.
>
> Minor Issue: Thanks for your suggestion; we have updated the usage of  “\citep{}” and "\cite{}" to improve readability.
> To ensure reproducibility, our code is available at https://anonymous.4open.science/r/predict-and-search-0F6F
>
> [1] Vinod Nair, Sergey Bartunov, Felix Gimeno, Ingrid von Glehn, Pawel Lichocki, Ivan Lobov, Brendan O’Donoghue, Nicolas Sonnerat, Christian Tjandraatmadja, Pengming Wang, Ravichandra Addanki, Tharindi Hapuarachchi, Thomas Keck, James Keeling, Pushmeet Kohli, Ira Ktena, Yujia Li, Oriol Vinyals, and Yori Zwols. Solving mixed integer programs using neural networks, 2020.

---

### Official Review · Reviewer_jKg3 · 2022-10-26

**Confidence:** 4
**Correctness:** 3
**Technical Novelty And Significance:** 3
**Empirical Novelty And Significance:** 2
**Recommendation:** 5

**Clarity, Quality, Novelty And Reproducibility:**

The clarity could be improved in my opinion by focusing more of the presentation on what is new to this work such as the trust region algorithm.


**Strength And Weaknesses:**

I think the paper considers an interesting problem, which is potentially of value to those working in the field of machine learning for combinatorial optimization. My specific comments and questions are as follows:

1) I think a more detailed comparison with the neural diving method of Nair et al 2020 would be really interesting. The authors mention the work of Nair et al 2020, however, it is not clear how the performance of their approach would compare.

2) In addition, I think it is interesting to consider how the performance of other neural network based MILP solver methods would compare against the proposed method.

3) The algorithm to solve Eq 11 seems critical for the proposed framework. In my opinion, it would increase the readability to include this algorithm in the main body of the paper.

4) Is the algorithm/solver for SOLVE(M’) in Alg. 1 discussed somewhere in the paper?

5) I think a simple yet interesting baseline to compare with the proposed method is by simply solving the relaxed LP and running the trust region algorithm on the solution of the relaxed LP.

6) As far as I understand, the reported numerical results are for the MILP instances in the test set, not training or validation set. Is this correct? Also, is this explicitly stated somewhere?

7) I think it would be nice to include a discussion on the neural network generalization. How does the performance on the training set compare to performance on the test set?

8) In my opinion, the claim that “Most of the end-to-end approaches directly predict solutions to MILP problems, ignoring feasibility requirements enforced by model constraints. As a result, the solutions provided by ML methods could potentially violate constraints.” needs references to some works where this is the case.

Minor issues:
1) The font size for Fig.3 is too small.
2) Typo in second sentence of the conclusion: supervise(d)-learning
3) Typo in page 8: “are took into consideration”
4) Typo in page 7, equation (6) and (6)
5) Extra parenthesis in the second line of the equation block for L(\theta): logp_(
6) Typo in the line under eq 11, \hat{x}_I - \hat{x}_I


**Summary Of The Paper:**

This paper considers a neural network based algorithm for solving MILPs. The main novelty lies in the proposed trust region algorithm to construct solutions from the output of the neural network.


**Summary Of The Review:**

I think the paper is interesting, however comparison to other work seems could be improved.

---

> ### Author Response · Authors · 2022-11-19
> **Official Response to Reviewer jKg3 part 1**
>
> Thanks very much for your insightful comments and suggestions! We have updated our paper based on the comments of the reviewers. Summary of revision:
> - We have added computational studies to compare our proposed approach against the neural diving method.
> - To improve the clarity of our paper, we have adjusted the paper's structure and added more references, as suggested.
>
> > **Q1**: I think a more detailed comparison with the neural diving method of Nair et al 2020 would be really interesting. The authors mention the work of Nair et al 2020, however, it is not clear how the performance of their approach would compare.
>
> Thank you for your suggestion, we have added computational studies to compare our proposed approach against the neural diving method. Please check the Appendix section IV. The computational results demonstrate that our approach performs better.
>
> >**Q2**: In addition, I think it is interesting to consider how the performance of other neural network based MILP solver methods would compare against the proposed method.
>
> In this work, we focus on developing learning-based algorithms for identifying high-quality feasible solutions to MILPs. To the best of our knowledge,  the only works that fall into this category are the Neural Diving method[1] and Neural Diving+Fixing Strategy [2].
>
> >**Q3**: The algorithm to solve Eq 11 seems critical for the proposed framework. In my opinion, it would increase the readability to include this algorithm in the main body of the paper.
>
> Thanks for your suggestion. We have moved Alg.1 to the main body of the paper.
>
> >**Q4**: Is the algorithm/solver for SOLVE(M’) in Alg. 1 discussed somewhere in the paper?
>
> Thanks for pointing out this. We have made updates; please check the end of section 3.2.2.
>
> >**Q5**: I think a simple yet interesting baseline to compare with the proposed method is by simply solving the relaxed LP and running the trust region algorithm on the solution of the relaxed LP.
>
> Thanks for raising this point.
> As suggested, we have conducted the experiments by solving relaxed LP and running the trust region algorithm with the solution of the relaxed LP.
> For the sake of fairness, experiments conducted on the same dataset will share the same set of parameters.
> Under such an experiment setting, the solution to relaxed LP failed to recognize even feasible solutions to the original problem.
>
> It is obvious from the results that, for most datasets, solutions to LP relaxations will lead to low-quality starting points and result in infeasible sub-problems; only on the CA dataset, using LP relaxation can produce solutions comparable to those given by the predict-and-search framework.
> In conclusion, the prediction part does contribute to finding high-quality starting points.
>
> The table below shows the computational results comparing our framework against using the solution to LP relaxation as the guidance for the search algorithm.
>
>
> | dataset | LP % feasible | LP avg. obj |  PS+SCIP % feasible |  PS+SCIP avg. obj |
> |---------|---------------|-------------|---------------|-------------|
> | IP      | 0             | –           | 100           | 15.46       |
> | WA      | 0             | –           | 100           | 702.35      |
> | IS      | 0             | –           | 100           | 685.03      |
> | CA      | 100           | 23,540.99   | 100           | 23,536.95   |
>
> >**Q6**: As far as I understand, the reported numerical results are for the MILP instances in the test set, not training or validation set. Is this correct? Also, is this explicitly stated somewhere?
>
> Yes, all reported numerical results are obtained with the test set. For clarity, we clarify this in the updated paper. Please check section 4.
>
> [1] Vinod Nair, Sergey Bartunov, Felix Gimeno, Ingrid von Glehn, Pawel Lichocki, Ivan Lobov, Brendan O’Donoghue, Nicolas Sonnerat, Christian Tjandraatmadja, Pengming Wang, Ravichandra Addanki, Tharindi Hapuarachchi, Thomas Keck, James Keeling, Pushmeet Kohli, Ira Ktena, Yujia Li, Oriol Vinyals, and Yori Zwols. Solving mixed integer programs using neural networks, 2020.
>
> [2]Taehyun Yoon. Confidence threshold neural diving, 2022.

---

> ### Author Response · Authors · 2022-11-19
> **Official Response to Reviewer jKg3 part 2**
>
> >**Q7**: I think it would be nice to include a discussion on the neural network generalization. How does the performance on the training set compare to performance on the test set?
>
> Thanks for the suggestion!  According to your suggestion, we conduct experiments of testing the performance of our method on the test set. As in the previous experiments, we use gap_p( performance gap ) as our metric. We obtained 40% and 37% ( Excluding IS ) performance improvements on the training and test set, respectively. The 3% performance difference indicates that our approach generalizes reasonably well. The table below shows the details.
>
> | dataset | bks      | SCIP obj  | SCIP gap  | PS+SCIP obj      | PS+SCIP gap | gap_p |
> |---------|----------|----------|------|----------|-------|-------|
> | IP      | 13.14    | 19.64    | 6.49 | 16.55    | 3.41  | 0.47  |
> | WA      | 707.00   | 710.20   | 3.20 | 709.00   | 2.00  | 0.375 |
> | IS      | 685.30   | 685.30   | 0    | 685.30   | 0     | --    |
> | CA      | 23599.14 | 23595.57 | 3.57 | 23596.85 | 2.29  | 0.36  |
> | AVG     |          |          |      |          |       | 0.40  |
>
> >**Q8**: In my opinion, the claim that “Most of the end-to-end approaches directly predict solutions to MILP problems, ignoring feasibility requirements enforced by model constraints. As a result, the solutions provided by ML methods could potentially violate constraints.” needs references to some works where this is the case.
>
> Thanks for the suggestion! We have cited a few references to support our claim.
>
> Minor issues:  Thank you!  We have fixed those typos as you pointed out and adjusted the font size of Figs.
>
> [1] Vinod Nair, Sergey Bartunov, Felix Gimeno, Ingrid von Glehn, Pawel Lichocki, Ivan Lobov, Brendan O’Donoghue, Nicolas Sonnerat, Christian Tjandraatmadja, Pengming Wang, Ravichandra Addanki, Tharindi Hapuarachchi, Thomas Keck, James Keeling, Pushmeet Kohli, Ira Ktena, Yujia Li, Oriol Vinyals, and Yori Zwols. Solving mixed integer programs using neural networks, 2020.
>
> [2]Taehyun Yoon. Confidence threshold neural diving, 2022.

---

### Author Response · Authors · 2022-11-19
**General response to all reviewers and the new revision**

We sincerely thank all the reviewers for their feedback and constructive comments.
Below we would like to highlight some new results and changes to the paper:
- We conducted additional computational studies to compare our proposed approach against the Neural Diving method[1], and the computational results demonstrate that our approach performs better. (See Appendix Section IV). Note that we did not incorporate the comparison in the previous submission for the following consideration: the work of Nair et al.[1] has not yet been accepted by any academic journals or conference proceedings. We have included a detailed comparison in the updated version of our paper.
- For reproducibility, we have now released the code of the proposed approach at https://anonymous.4open.science/r/predict-and-search-0F6F
- We have corrected typos and improved the readability in the newly uploaded version.

[1] Vinod Nair, Sergey Bartunov, Felix Gimeno, Ingrid von Glehn, Pawel Lichocki, Ivan Lobov, Brendan O’Donoghue, Nicolas Sonnerat, Christian Tjandraatmadja, Pengming Wang, Ravichandra Addanki, Tharindi Hapuarachchi, Thomas Keck, James Keeling, Pushmeet Kohli, Ira Ktena, Yujia Li, Oriol Vinyals, and Yori Zwols. Solving mixed integer programs using neural networks, 2020.

---

### Decision · Program_Chairs · 2023-01-20

**Decision:**

Accept: poster

**Justification For Why Not Higher Score:**

Novelty is sufficient over ND, but not to spotlight level.

**Justification For Why Not Lower Score:**

The post-rebuttal comparisons prove the contribution of the paper.

**Metareview: Summary, Strengths And Weaknesses:**

All reviewers find the paper's proposal appealing (from the most negative review: "It is conceivable predicted variables from ML model is often not feasible, hence making only a "soft" commitment and searching in an L1 ball seems like a practically useful solution. "), and all are satisfied that the novelty over Neural Diving (ND) of Nair et al is sufficient.

The original paper had not compared against ND, but post-rebuttal, these comparisons are included, and show the advantages of the paper's proposal over the existing work.

Reviewers did not update their scores post rebuttal, but the paper has been carefully considered by the AC and SAC, and discussed with the PC chairs, and it is clear that the post-rebuttal paper is significantly stronger than the original paper, to which the numerical scores applied.  Hence, the text of the reviews, combined with the post-rebuttal improvements to the paper, supercedes the raw numerical scores in accepting this paper.

For final copy, the comparison with ND should be moved from the appendix to appear prominently in the main paper---all reviewers (and the authors) agree that ND is closely related work.  The authors reasoning that ND "has not yet been accepted by any academic journals or conference proceedings" is understandable, in the view of the AC and SAC, given the paper's 100 citations, it is /de facto/ an academic publication.


**Note From Pc:**

if the above contains the word "oral" or "spotlight" please see: "oral" presentation means -> notable-top-5% and "spotlight" means -> notable-top-25%. As stated in our emails, we are disassociating presentation type from AC recommendations